# Evaluation of Different Activity of *Lactobacillus* spp. against Two *Proteus mirabilis* Isolated Clinical Strains in Different Anatomical Sites In Vitro: An Explorative Study to Improve the Therapeutic Approach

**DOI:** 10.3390/microorganisms11092201

**Published:** 2023-08-31

**Authors:** Alessandra Fusco, Vittoria Savio, Adriana Chiaromonte, Alberto Alfano, Sergio D’Ambrosio, Donatella Cimini, Giovanna Donnarumma

**Affiliations:** Department of Experimental Medicine, University of Campania “Luigi Vanvitelli”, 80138 Naples, Italy; vittoriasavio@libero.it (V.S.); adrianachiaromonte1993@gmail.com (A.C.); alberto.alfano@unicampania.it (A.A.); donatella.cimini@unicampania.it (D.C.)

**Keywords:** UTIs, prostatitis, *Proteus mirabilis*, *Lactobacillus* spp., cytokines, biofilm

## Abstract

Urinary tract infections (UTIs) and catheter-associated UTIs (CAUTIs) are the principal hospital-acquired infections. Between these, bacterial prostatitis is believed to be the leading cause of recurrent UTIs in men under 50 years of age and is often unresponsive to antibiotic treatment. *Proteus mirabilis* is more commonly associated with UTIs in these abnormalities, especially in patients undergoing catheterization. *Lactobacillus* spp. are an important component of the human microbiota and occur in large quantities in foods. Probiotics are proposed as an alternative to antibiotic therapy in the treatment of urinary tract infections. In addition to their ability to produce antimicrobial metabolites, they have immunomodulatory activity and do not cause side effects. For this reason, the combination of probiotic microorganisms and conventional drugs was considered. The aim of this work was to select the most active *Lactobacillus* strains against two clinical isolates of *P. mirabilis* on bladder and prostatic epithelium, potentially exploitable to improve the clinical management of UTIs.

## 1. Introduction

Urinary tract infections (UTIs) represent one of the most frequent bacterial infections, with an onset of approximately 150 million cases per year worldwide [1], affecting newborns and men and women of all ages. They can cause serious complications, including premature birth, infant kidney damage, relapses, pyelonephritis with sepsis, and complications due to frequent use of antibiotic drugs [2,3]. UTIs are clinically classified into complicated and uncomplicated UTIs; the former usually affect patients without structural or neurological urinary tract abnormalities [2,3], while complicated UTIs are associated with urinary tract risk factors or pre-existing host immune factors, including urinary retention due to neurological disease, renal insufficiency, urinary obstruction, immunosuppression, pregnancy, kidney transplantation, the presence of stones, and indwelling drainage devices or catheters [4].

In men, one of the most frequent complications of UTIs is represented by the implication of the prostate as a parenchymatous organ. Prostatitis is a very common syndrome, observed in over 90% of febrile infections [5,6,7], characterized by voiding problems (obstructive or irritative) and pain (pelvic, genitourinary, or rectal) and sometimes associated with sexual dysfunction (ejaculatory discomfort and hematospermia), and frequent recurrences [8], conditions that greatly compromise quality of life. The appearance of recurrences is linked to the inability of antibiotics to penetrate the prostatic epithelium and consequently to reach sufficient concentrations to carry out their bactericidal activity.

One of the main aetiological agents associated with the onset of complicated UTIs and catheter-associated UTIs (CAUTIs) is *Proteus mirabilis*, a mobile rod-shaped Gram-negative bacterium belonging to the *Enterobacteriaceae* family, originally discovered in 1885 by Hauser. *P. mirabilis* is widely distributed in the environment, mainly in water and soil, where it plays an important role in the decomposition of organic material of animal origin [9,10], and colonizes human and animal gastrointestinal tracts [11,12].

One of the main characteristics of *P. mirabilis* is dimorphism. This phenomenon begins when the bacteria come into contact with a solid surface and undergo a phenotypic switch from a shorter “swim cell” to elongated “swarm cell” morphology. Massive expression of most of *P. mirabilis*’s virulence factors, such as enzyme urease, IgA protease, and hemolysin have been observed to be associated with the swarming phenotype [13,14]. It has also been shown that swarming cells invade uroepithelial cells more effectively than vegetative cells [15].

Kidney damage and the formation of kidney and bladder stones (urolithiasis) further complicate UTIs and CAUTIs caused by *P. mirabilis*. In fact, the enzyme urease acts by hydrolyzing urea into ammonia and carbon dioxide, with consequent alkalization of the urine and precipitation of crystals of struvite (magnesium ammonium phosphate) and apatite (calcium hydroxyl phosphate) [16]. These crystals form a crystalline biofilm which is deposited on the catheters, encrusting them and blocking the flow of urine [14,17]. This blockage consequently causes urine retention in the bladder, which favors the development of serious ascending kidney infections (pyelonephritis), with serious complications such as septicemia and shock. The biofilm also contributes to the chronicity of the infection, as it reduces the penetration of antibiotics and protects the bacteria from the host immune response [14,16,17].

The ever-increasing expansion of antibiotic resistance phenomena is making conventional therapies unable to cope with infections and requires an urgent development of alternative therapeutic strategies. In the case of *P. mirabilis*, antibiotic resistance was found in 48% of clinically isolated strains [18,19].

One of the main approaches in recent years is the indirect modulation of the intestinal microbial community through the administration of live beneficial microorganisms, the so-called probiotics [19,20]. The most common group of bacteria used as probiotics belongs to the lactic acid bacteria group, mainly *Lactobacillus* spp. [21]. *Lactobacillus* strains are able to improve digestion, absorption, and nutrient availability in both farm animals and humans [22,23], and have shown promise in ameliorating many disease states [24,25,26,27,28,29]. They are also frequently endowed with antibiotic resistance, which is useful in the case of combination therapies with antibiotics [22,23].

In recent years, we have characterized two clinically isolated strains of *P. mirabilis*, which we named PM1 and PM2, highlighting the differences in their pathogenetic mechanisms [5,30].

It has been amply demonstrated that the use of lactobacilli can improve the course of UTIs [31,32,33]; however, in choosing the specific strains to use, the diversity of behavior of the infecting pathogens and the response of the epithelium towards which the therapy is directed must be taken into consideration.

For all these reasons, the purpose of this current work was to evaluate in vitro the effect of different strains of *Lactobacillus* spp. during infections caused by PM1 and PM2 in the bladder and prostatic epithelial cells, to select those with the best efficacy both for each pathogenic strain and for each epithelium.

## 2. Materials and Methods

### 2.1. Bacterial Strains

The *P. mirabilis* strains (PM1 and PM2) were isolated in 2009 from patients with infectious diseases of the renal system at the Bacteriology Division of the Microbiology and Virology Unit, University Hospital of Campania Luigi Vanvitelli. These strains were first identified by the API20NE identification system (BioMèrieux, Grenoble, France), and then confirmed by partial DNA sequencing of the 16S-23S rRNA gene using primers wl-5793 (5′-TGT ACA CAC CGC CCG TC-3′) and wl-5794 (5′-GGT ACT TAG ATG TTT CAG TTC-3′) and finally by BLAST analysis [30]. The two strains PM1 and PM2 have also been previously characterized for their swarming activity, for their ability to form biofilms and to induce the production of proinflammatory cytokines and antimicrobial peptides in the host, for the different expression of their virulence characters (*WosA*, *FlhDC*, *ZapA*), for their ability to induce the activation of apoptotic pathways, for the different antibiotic resistance, and their molecular profile has been investigated through a genetic fingerprinting with RAPD-PCR [5,30].

All strains were grown in Luria–Bertani medium (LB-Oxoid, Milan, Italy) at 37 °C in aerobic conditions.

*Lactobacillus* strains isolated from buffalo milk, including *Lacticaseibacillus paracasei* IMC 502, *Lacticaseibacillus rhamnosus* IMC 502, *Bifidobacterium lactis* HN019™, *Levilactobacillus brevis* SP-48 [34], and ATCC strains *Lactobacillus acidophilus* ATCC 4356™ and *Lactiplantibacillus plantarum* ATCC 8014™ were grown in Man, Rogosa and Sharpe broth (MRS-Oxoid, Milan, Italy) at 37 °C in microaerophilic conditions for 24 h.

### 2.2. Biofilm Formation Assay

For the biofilm assay, overnight cultures of PM1 and PM2 were diluted to obtain a concentration of 10^7^ CFUs/mL, and aliquots (200 μL) of the diluted bacterial suspension were placed into 96-well, flat-bottomed, sterile polystyrene microplates (Costar, Corning, Inc., Corning, NY, USA), with or without *Lactobacillus* spp. at the same concentration, and incubated overnight at 37 °C. The biofilm formed was quantified by a modification of the crystal violet assay [35]. After 24 h, the attached bacteria were washed twice with 200 μL of PBS (Microgem, Naples, Italy) and air-dried for 45 min. The wells were then stained with 200 μL of 1% aqueous crystal violet solution (Sigma-Aldrich, Merck, Milan, Italy) for 45 min. The plates were rinsed with 200 μL of sterile distilled water to remove excess dye and air-dried. The dye associated with the attached biofilm was dissolved in a solution of 200 μL of 100% ethanol, and the OD_570/655_ absorbance was measured on a microplate reader (Biorad, Hercules, CA, USA).

### 2.3. Counting of Sessile Bacteria

The biofilm of PM1 and PM2 with or without lactobacilli, formed as previously described, was rinsed twice with PBS, then lysed with a solution of 0.1% Triton X-100 (Merck, Milan, Italy). Aliquots of cell lysates were serially diluted and plated on Hektoen enteric agar (Oxoid, Milan, Italy) and incubated at 37 °C overnight to quantify sessile viable bacteria (CFUs/mL).

### 2.4. Cells Cultures and Treatment

PC3 (human prostatic adenocarcinoma, ATCC CRL-1435™) and T24 (epithelioid carcinoma of the human urinary bladder, ATCC HTB-4™) cells were cultured respectively in RPMI and McCoy’s 5a Medium (Gibco, Merck, Milan, Italy) supplemented with 1% Penstrep, 1% glutamine, and 10% fetal calf serum (Invitrogen, Carlsbad, CA, USA) at 37 °C in air and 5% CO_2_. Subsequently, cells were dispensed into 6-well plates and left to grow until 80% of confluence.

For lactobacilli screening, semiconfluent monolayers were infected with 10^8^ CFUs/mL of different *Lactobacillus* strains at a multiplicity of infection (MOI) of 100 [36,37,38] for 6 h at room temperature at 37 °C. For evaluation of activity of probiotics against *P. mirabilis*, cells were infected with exponentially growing PM1 and PM2 at an MOI of 100 bacteria/cell, with or without selected *Lactobacillus* strains. Infection was carried out for 6 (for gene expression analysis) and 24 h (for ELISA assay) at 37 °C in 5% CO_2_.

### 2.5. Analysis of Flagellar Gene Expression

Supernatant collected after 6 h of infection with PM1 and PM2 with or without selected *Lactobacillus* strains, were used to extract total bacterial RNA with Tri-Reagent^®^ (Sigma-Aldrich, Merck, Milan, Italy), as per manufacturer’s instructions. The expression of the *FlhDC* gene (Table 1) was evaluated by real-time PCR as shown below.

### 2.6. Real-Time PCR

At the end of each experiment, cells were washed three times with sterile PBS, and the total RNA was extracted using High Pure RNA Isolation Kit (Roche Diagnostics, Monza, Italy).

A total of 200 ng of cellular RNA were reverse-transcribed (Expand Reverse Transcriptase, Roche, Monza, Italy) into complementary DNA (cDNA) using random hexamer primers (random hexamers, Roche, Monza, Italy) at 42 °C for 45 min, according to the manufacturer’s instructions. Real-time PCR for IL-6, IL-8, TNF-α, IL-1α, and HBD-2 was carried out with the LC Fast Start DNA Master SYBR Green kit using 2 µL of cDNA, corresponding to 10 ng of total RNA in a 20 μL final volume, 3 mM MgCl_2_, and 0.5 μM sense and antisense primers (Table 1). After amplification, melting curve analysis was performed by heating to 95 °C for 15 s at a temperature transition rate of 20 °C/s, cooling to 60 °C for 15 s with a temperature transition rate of 20 °C/s, and then heating the sample at 0.1 °C/s to 95 °C. The results were then analyzed using LightCycler^®^ 2.0 software (Roche Diagnostics, Monza, Italy). The standard curve of each primer pair was established with serial dilutions of cDNA. All PCRs were run in triplicate. The specificity of the amplification products was verified using electrophoresis on a 2% agarose gel and visualization by ethidium bromide staining [30].

### 2.7. Bacterial Internalization Assay

T24 and PC3 cells were infected with PM1 or PM2 with or without selected *Lactobacillus* spp. strains, as previously described. After 3 h of incubation at 37 °C, cell supernatants were collected and infected. Monolayers were extensively washed with sterile PBS and further incubated for another 2 h in the RPMI or McCoy’s medium, supplemented with 250 μg/mL gentamicin sulphate (Sigma-Aldrich, Merck, Milan, Italy) to kill the extracellular bacteria. At the end of the experiments, infected monolayers were extensively washed in PBS, then lysed with a solution of 0.1% Triton X-100 (Sigma-Aldrich, Merck, Milan, Italy) in PBS for 10 min at room temperature to count internalized bacteria [39]. Aliquots of cell lysates were serially diluted and plated on Hektoen enteric agar and incubated at 37 °C overnight to quantify viable intracellular bacteria (CFUs/mL). The efficiency was calculated according to the following formula:$$\%\;efficiency\;of\;internalization=\frac{number\;of\;internalized\;bacteria}{total\;number\;of\;bacteria}\times 100$$

### 2.8. ELISA Assay

The presence of IL-6, IL-8, IL-1α, TNF-α, and HBD-2 in cellular supernatants was analyzed using enzyme-linked immunosorbent assay (ELISA; Elabscience Biotechnology Inc., Houston, Texas, USA; Phoenix Pharmaceuticals, Inc., Burlingame, CA 94010, USA). Briefly, the cell culture supernatants were collected and centrifuged at 1000× *g* for 20 min to eliminate cell debris. Then, 100 μL conditioned medium was added to the appropriate wells and incubated at 37 °C for 90 min. Thereafter, the liquid was decanted, the biotinylated detection antibody working solution was added to the wells, and the mixture was incubated at 37 °C for 60 min. The wells were washed 3 times with wash buffer, the HRP conjugate working solution was added, and the wells were incubated at 37 °C for 30 min. Then, the wells were washed 5 times with wash buffer, and the substrate reagent was added for 15 min. The stop solution was added to the wells, and the OD values were determined at 450 nm [40].

### 2.9. Statistical Analysis

Significant differences among groups were assessed through two-way ANOVA by using GraphPad Prism 9.0, and the comparison between the means was calculated using Student’s *t*-test. The data are expressed as the means ± standard deviation (SD) of three independent experiments.

## 3. Results

### 3.1. Antibiofilm activity of Lactobacillus spp. against P. mirabilis

The biofilm formation assays of PM1 and PM2 were carried out in the presence of the different strains of *Lactobacillus*, and at the end of the incubation, the biofilm was quantified by spectrophotometric reading following crystal violet staining. Due to the inherent adhesive ability of lactobacilli, the results obtained (Figure 1a) show that there is apparently no significant difference in the amount of biofilm formed in the presence or absence of *Lactobacillus*, as the values obtained were the result of the presence of polymicrobial biofilm composed of both *Proteus* and probiotic strains. For this reason, the biofilm was lysed with 0.1% Triton X-100 and serial dilutions were plated on Hektoen agar, a selective medium for enterobacteria, which would ensure the growth of PM1 and PM2 species but not of lactobacilli, to count the number of effective viable sessile cells of pathogenic species.

The data obtained (Figure 1b) confirmed that the number of CFUs/mL of PM1 and PM2 decreases significantly in the presence of the different strains of *Lactobacillus*, with the exception of *L. acidophilus*, which does not seem to have the ability to contrast the formation of biofilms by PM1.

### 3.2. Selection of Best Lactobacillus spp. Strain for Each Epithelium

PC3 and T24 cells were treated for 6 h with the different strains of lactobacilli, and after this time, the strains were tested for their immunomodulatory activity by analyzing the gene expression of *IL-6*, *IL-1α*, and *HBD-2*. Our data demonstrate that for PC3, only two strains, *L. paracasei* and *L. rhamnosus*, are able to significatively downregulate the expression of pro-inflammatory cytokines and to upregulate the expression of HBD-2, whereas for T24, all strains analyzed have shown anti-inflammatory and immunomodulatory activity (Figure 2a,b).

### 3.3. Analysis of Lactobacillus spp. Anti-Inflammatory Activity against Different P. mirabilis Strains Clinically Isolated in Different Epithelia

PC3 and T24 cells were treated with PM1 and PM2 for 6 h alone or in the presence of the lactobacilli strains selected for the best anti-inflammatory and immunomodulatory activity for each epithelium. The results obtained also show in this case extremely different behavior of the lactobacilli in the two different epithelia. In fact, while in the prostatic epithelium, both strains of lactobacilli are active almost exclusively on PM2, both in terms of anti-inflammatory capacity and in the production of HBD-2 (Figure 3). In the bladder epithelium, it seems that the main activity is exerted against PM1, with the sole exception of IL-8 and HBD-2 (Figure 4). The protein dosage performed by ELISA confirms the data from the molecular analyses.

### 3.4. Inhibition of P. mirabilis Invasiveness

The analysis of the invasiveness inhibition of PM1 and PM2 by lactobacilli also shows in this case a diversified action based on the type of epithelium. In fact, as far as the prostatic epithelium is concerned (Figure 5a), the probiotic strains only act significantly on PM2, while on the bladder epithelium, all the analyzed strains of lactobacilli were shown to be able to significantly inhibit the invasive ability both of PM1 and PM2 (Figure 5b).

## 4. Discussion

*P. mirabilis* is often associated with the onset of UTIs, which lead to cystitis, urethritis, chronic inflammation, acute pyelonephritis, and bacteremia [2,5,30]. The incidence of UTIs is very high in hospitalized patients and among elderly patients residing in care facilities [41], especially in men over 60 years of age. One of the most common consequences of recurrent UTIs in men is bacterial prostatitis [5]. In this syndrome, recurring episodes of bladder bacteriuria can cause bacteria to stagnate and accumulate in the prostate, resulting in the development of infection. Bacterial prostatitis has an incidence in the community of about 0.9–2 cases per 1000 men under the age of 55 and 7.7 cases per 1000 men aged 85 and over [42], and its onset is favored by the presence of prostatic dysfunction, a disorder that frequently occurs with advancing age, the most common symptom of which is benign prostatic hyperplasia (BPH).

The main problem in the treatment of patients affected by this type of infection is the frequent onset of antibiotic multi-resistance phenomena, now increasingly common in strains isolated in hospital settings [43], for which it is of fundamental importance to obtain a correct diagnosis as quickly as possible in order to set up an effective and adequate therapy. Moreover, for *P. mirabilis*, multidrug resistance (MDR) in clinical isolates is leading to a serious public health problem for hospitalized patients [44,45,46]. As a result, the use of alternative therapies based on probiotics, in particular lactobacilli, is developing more and more both in the medical field and in agriculture [20,47,48]. The intake of probiotics is understood as the application of live microorganisms in order to obtain a health benefit, both in the prevention of various diseases and in the improvement of the general conditions of the host organism [24,25,26,27,28,29,49].

It has been widely reported [50,51,52] that the use of lactobacilli in UTIs has several advantages. It seems in fact that, although the mechanisms of action have not yet been fully clarified, the strains of lactobacilli can act at least in three different ways [52]. First, they can exert a bacteriostatic activity due to an effect of direct competition with the uropathogens in terms of subtraction of nutrients and binding to the attachment sites [53]. Second, they can downregulate the expression of uropathogens’ virulence genes thanks to the production of metabolic by-products (such as lactic acid and hydrogen peroxide) [33]. An example of this mechanism was demonstrated in an in vitro study in which *Lactobacillus* spp. by-products inhibited the expression of genes coding for type 1 and P fimbriae in *Escherichia coli*, inhibiting its adhesive and invasive properties [33]. Third, they can exert a bactericidal activity on uropathogens through the production of bacteriocins, antimicrobial peptides that act in a strain-specific way [54].

In this paper, we analyzed the effect in vitro of different *Lactobacillus* strains, in particular, *L. paracasei*, *L. rhamnosus*, *B. lactis*, *L. brevis*, *L. acidophilus,* and *L. plantarum,* during infections sustained by two clinical isolated strains of *P. mirabilis* in bladder and prostatic epithelium.

For this purpose, we performed first a screening to evaluate which *Lactobacillus* strain was more suitable according to the type of epithelium. We then treated PC3 and T24 cells for 6 h with the different *Lactobacillus* strains, and subsequently we evaluated the immunomodulatory activity by analyzing the cellular expression of IL-6, IL-1α and of the antimicrobial peptide HBD-2. The results obtained from these preliminary experiments showed us that the two different epithelia respond in a totally different way to the treatment with lactobacilli. In fact, after 6 h of treatment, in the PC3 cells only, *L. paracasei* and *L. rhamnosus* were endowed with anti-inflammatory activity and the ability to induce the production of HBD-2, while in the T24 cells, the anti-inflammatory and immunomodulatory effect was obtained by all strains analyzed. On the basis of these preliminary data, it was therefore decided to proceed with the subsequent experiments using only the strains that had shown immunomodulatory activity.

Subsequently, PC3 and T24 cells were infected for 6 h with two isolated clinical strains of *P. mirabilis*, respectively named PM1 and PM2, already studied by our group for their different characteristics [5,30], in the presence and absence of probiotics, and the ability of the latter to reduce inflammation notoriously induced by *P. mirabilis* [30,55] and to activate the immune defenses were evaluated. The modulation of the expression of the pro-inflammatory cytokines IL-6, IL-8, IL-1 a, and TNF-a and of HBD-2 was then analyzed. The data obtained showed a totally different behavior to the strains analyzed with respect to the type of epithelium and the type of clinical isolate of *P. mirabilis*. In fact, in the prostatic epithelium, both lactobacilli tested showed activity only against PM2. The reverse occurred in the bladder epithelium, where, except for IL-8 and HBD-2, most of the anti-inflammatory activity was exerted by all probiotic strains, mainly on PM1.

*P. mirabilis* is known for its ability to swarm on agar plates, due to a cyclic differentiation process from vegetative cell to swarming cell [17], which is triggered when the vegetative cell comes into contact with a solid surface. Once differentiated, the swarming cell is 10 to 40 times longer than the vegetative cell and has a significantly increased number of flagella. This phenotypic shift is associated with changes in peptidoglycan, lipopolysaccharide (LPS), and membrane fatty acid composition [14,56]. The elongated swarming cells, thanks to cell–cell signaling, line up in multicellular “rafts” and exit the original site of inoculation in a highly coordinated way that depends on their interactions and on an uncharacterized slime and a capsular polysaccharide named “colony migration factor” that enclose them and facilitate their movement [14]. Simultaneously, urease alkalizes urine by splitting urea into ammonia and bicarbonate, resulting in the precipitation of assembled polyvalent ions [17]. In this microenvironment, bacteria multiply, forming microcolonies which can mature into so-called crystalline biofilms which protect them from the action of antibiotics and give them the ability to evade the immune response [17,30,57].

The swarming process is cyclically interrupted, and the cells return to their vegetative form via a process called consolidation. This cycle, repeating itself over and over again, gives rise to the appearance of the characteristic concentric rings present on the agar plates.

In a second part of the experiments, the ability of selected probiotics to interfere with PM1 and PM2 biofilm formation was evaluated. The results obtained indicated that all strains tested have the ability to significantly decrease the biofilm formation of both clinical isolates of *P. mirabilis*, probably acting with a competition mechanism for substrate binding. This mechanism is explained as the quantification of the formed biofilm, obtained by spectrophotometric reading following a crystal violet assay, and does not show a decrease in the amount of biofilm, probably since the lactobacilli (of which the adhesive properties are known) are also found to be adhered to the surface. On the other hand, the count of CFUs/mL on a selective medium shows a significant decrease in the number of colonies of PM1 and PM2 in the presence of lactobacilli compared to the controls.

Swarming activity is influenced by environmental conditions and by the activation of a large number of genes. Among these, the *flhDC* gene is a heterodimeric activator that plays a central role in the differentiation process, as it mediates the increase in flagellin expression by coupling its transcription to the differentiation of the swarm cells. Any synthetic process involving overexpression of *flhDC* results in premature differentiation, hyperflagellation, and increased velocity during swarming [58].

The cells supernatants infected with PM1 and PM2, with and without probiotics, were collected, and bacterial mRNA was extracted from them to analyze the influence of lactobacilli on *flhDC* gene expression. The results obtained show that in PC3 cells, the downregulation of *flhDC* occurs only in the presence of PM2, while in T24 cells all strains have been shown to significantly downregulate the expression of *flhDC,* both during PM1 and PM2 infection.

Finally, the ability of lactobacilli to reduce the invasion of PM1 and PM2 was evaluated by performing invasiveness assays and CFUs/mL counting. In this case, a diversity of responses from the cells was found. In fact, as for the *flhDC* gene, the invasiveness was also only significantly reduced for PM2 in PC3, and for both strains for T24.

## 5. Conclusions

Although conventional antibiotic-based therapies remain the elective choice in the treatment of recurrent UTIs, the continuous expansion of antibiotic resistance phenomena worldwide has made it necessary to develop new therapeutic strategies. Among these, the most promising ones would seem to be those based on the local application or oral intake of probiotics [59,60,61]. In addition to their ability to produce antimicrobial metabolites with an action comparable to that of antibiotics, they have immunomodulatory activity and do not cause side effects.

For this reason, the combination of probiotic microorganisms and conventional drugs was considered. The advantages of this synergism include an increase in the rate of eradication of microbial infections, faster healing times, halving the dose of conventional drug required, and a consequent reduction in the side effects caused by classical therapy [19,20,59].

In this study, we have demonstrated that the strains of *Lactobacillus* spp. may have a beneficial effect during *P. mirabilis* infections, but we have also shown that the effect is species-specific and conditioned both by the infecting strain and by the type of epithelium. It is clear that in vitro studies have some limitations as they are not affected by the influence of the pharmacological absorption and metabolization which takes place in vivo, but they can be valid bases which provide preliminary information to be investigated in clinical studies. Moreover, it still remains to be established which is the most effective type of administration to ensure effective reaching of anatomical sites such as the bladder and the prostate. In addition, it is necessary to perform in-depth studies to well investigate the interaction between probiotics, uropathogens, and the host immune system, since it is a misconception that any probiotic strain can be effective, as the cellular response can vary according to the site of action and the type of infecting strain. A better understanding of this mechanism will help direct future research on the topic and the formulation of specific and effective alternative therapies.

## Figures and Tables

**Figure 1 microorganisms-11-02201-f001:**
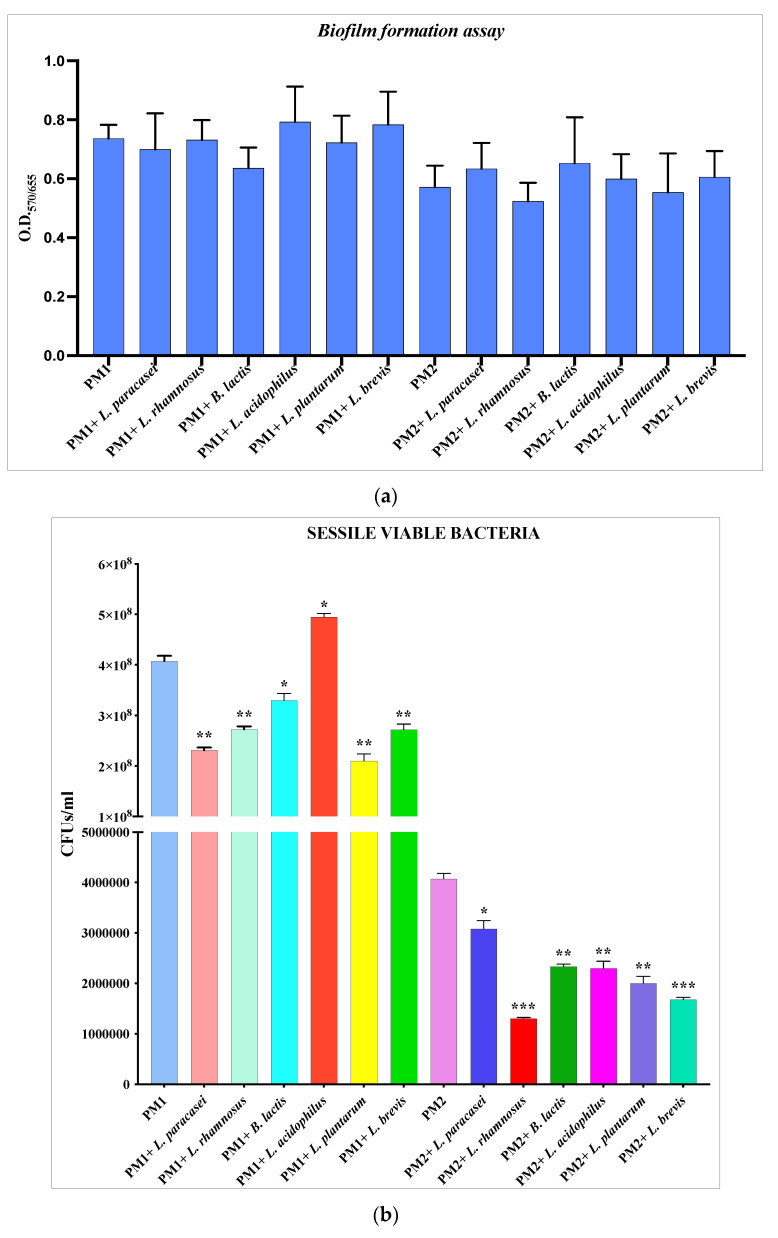
Biofilm formation assay. (**a**) Absorbance value reading at O.D._570/655_ after crystal violet staining. (**b**) Sessile viable bacteria counting. Data are representative of three different experiments ± SD. Significant differences are indicated by * *p* < 0.05, ** *p* < 0.01, *** *p* < 0.001.

**Figure 2 microorganisms-11-02201-f002:**
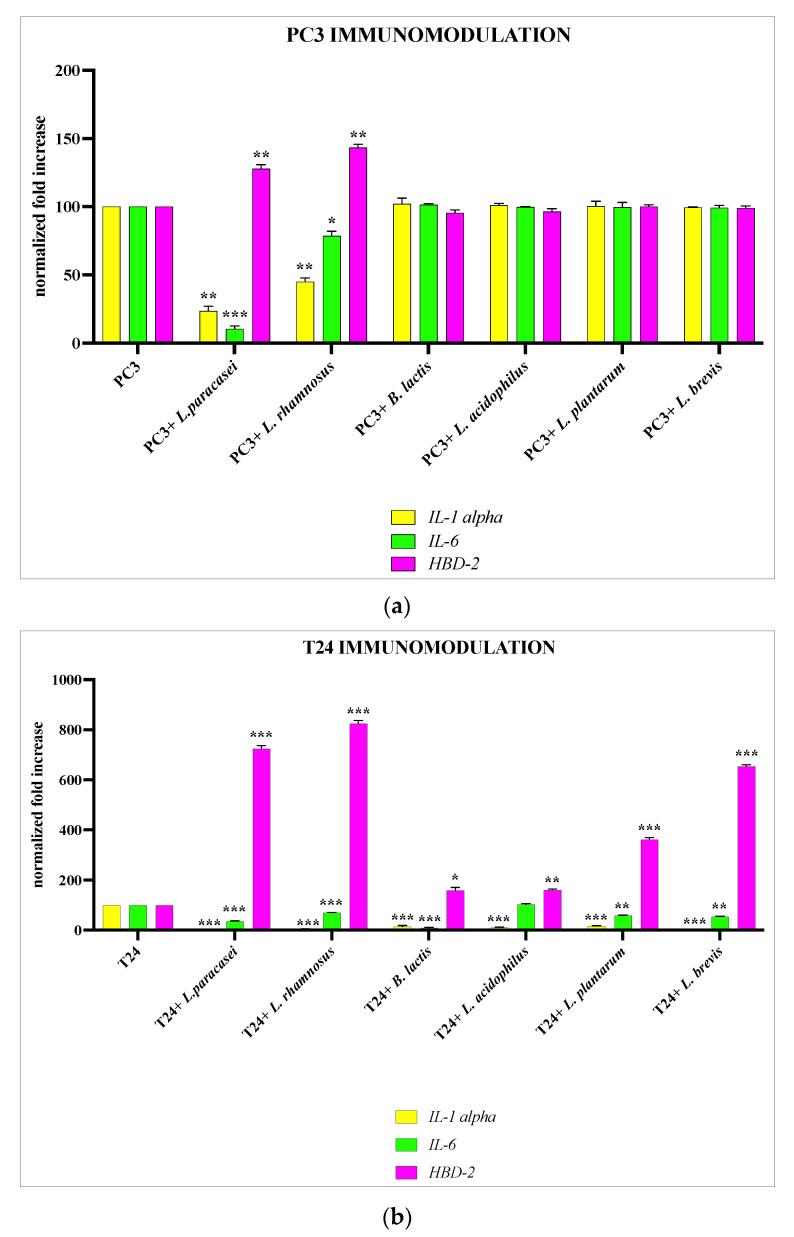
Immunomodulation. Comparison between gene expression of *IL-6*, *IL-1a*, and *HBD-2* in PC3 (**a**) and T24 (**b**) cells treated with different strains of *Lactobacillus* spp. Data are representative of three different experiments ± SD. Significant differences are indicated by * *p* < 0.05, ** *p* < 0.01, *** *p* < 0.001.

**Figure 3 microorganisms-11-02201-f003:**
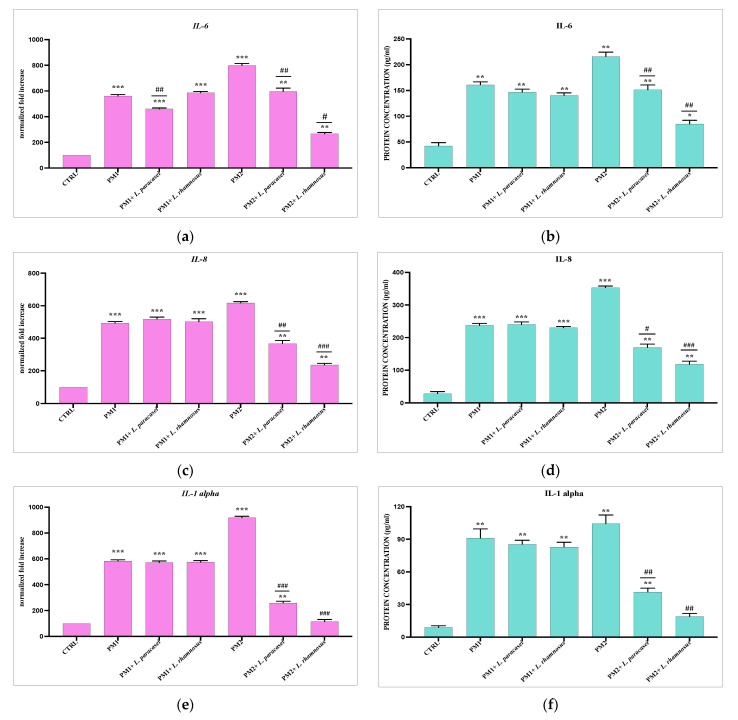
Comparison between relative gene expression (**a**,**c**,**e**,**g**,**i**) and protein concentration (**b**,**d**,**f**,**h**,**j**) in PC3 cells infected with PM1 and PM2 with or without *Lactobacillus* spp. Data are representative of three different experiments ± SD. Significant differences are indicated by * *p* < 0.05, ** *p* < 0.01, *** *p* < 0.001 for comparison with respect to untreated cells (CTRL) arbitrarily assigned to 100, or ^#^ *p* < 0.05, ^##^ *p* < 0.01, ^###^ *p* < 0.001 for comparison with respect to PM1 or PM2 alone.

**Figure 4 microorganisms-11-02201-f004:**
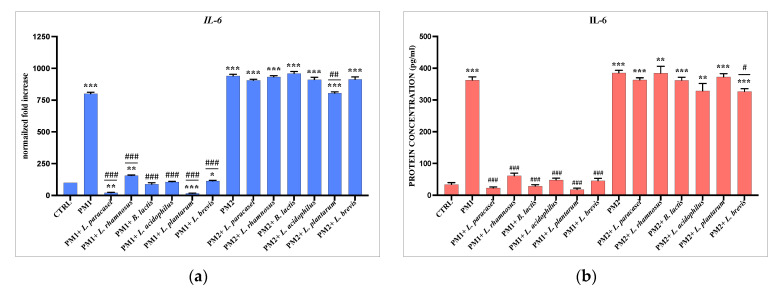
Comparison between relative gene expression (**a**,**c**,**e**,**g**,**i**) and protein concentration (**b**,**d**,**f**,**h**,**j**) in T24 cells infected with PM1 and PM2 with or without *Lactobacillus* spp. Data are representative of three different experiments ± SD. Significant differences are indicated by * *p* < 0.05, ** *p* < 0.01, *** *p* < 0.001 for comparison with respect to untreated cells (CTRL) arbitrarily assigned to 100, or ^#^ *p* < 0.05, ^##^ *p* < 0.01, ^###^ *p* < 0.001 for comparison with respect to PM1 or PM2 alone.

**Figure 5 microorganisms-11-02201-f005:**
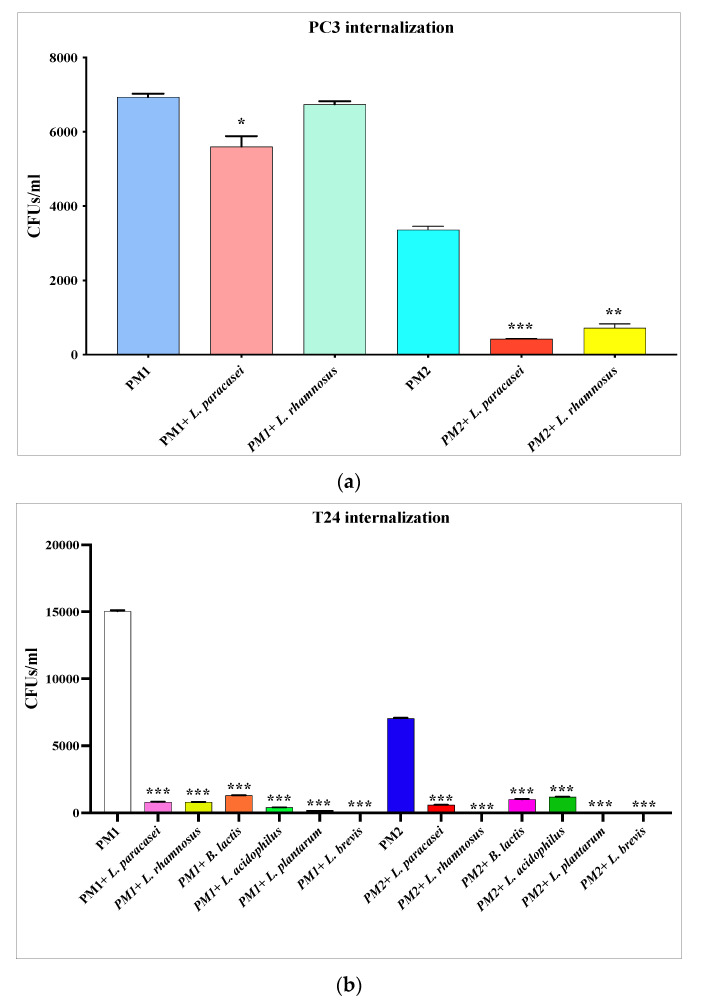
PM1 and PM2 invasion assays in PC3 (**a**) and T24 (**b**) cells alone or in the presence of *Lactobacillus* spp. Number of viable cell-associated bacteria was determined by host cell lysis, plating, and counting of CFUs/mL. Data are representative of three different experiments ± SD. Significant differences are 540 indicated by * *p* < 0.05, ** *p* < 0.01, *** *p* < 0.001.

**Table 1 microorganisms-11-02201-t001:** Primer sequences and amplification programs used in this work.

Gene	Primer Sequences	Conditions	Product Size (bp)
*IL-6*	5′-ATGAACTCCTTCTCCACAAGCGC-3′ 5′-GAAGAGCCCTCAGGCTGGACTG-3′	5″ at 95 °C, 13″ at 56 °C, 25″ at 72 °C for 40 cycles	628
*IL-8*	5′-ATGACTTCCAAGCTGGCCGTG-3′ 5′-TGAATTCTCAGCCCTCTTCAAAAACTTCTC-3′	5″ at 94 °C, 6″ at 55 °C, 12″ at 72 °C for 40 cycles	297
*IL-1a*	5′-GCATCCAGCTACGAATCTCC-3′ 5′-CCACATTCAGCACAGGACTC-3′	5″ at 95 °C, 14″ at 58 °C, 28″ at 72 °C for 40 cycles	708
*TNF-a*	5′-CAGAGGGAAGAGTTCCCCAG-3′ 5′-CCTTGGTCTGGTAGGAGACG-3′	5″ at 95 °C, 6″ at 57 °C, 13″ at 72 °C for 40 cycles	324
*HBD-2*	5′-GGATCCATGGGTATAGGCGATCCTGTTA-3′ 5′-AAGCTTCTCTGATGAGGGAGCCCTTTCT-3′	5″ at 94 °C, 6″ at 63 °C, 10″ at 72 °C for 50 cycles	198
*flhDC*	5′-CGCACATCAGCCTGCAAGT-3′ 5′-GCAGGATTGGCGGAAAGTT-3′	5″ at 94 °C, 6″ at 53 °C, 7″ at 72 °C for 40 cycles	90
*β-actin*	5′-GACGACGACAAGATAGCCTAGCAGCTATGAGGATC-3′ 5′-GAGGAGAAGCCCGGTTAACTTCCGCAGCATTTTGCGCCA-3′		243

## Data Availability

The authors confirm that the data supporting the findings of this study are available within the article.

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
