# Peer review of "Evaluation of Different Activity of Lactobacillus spp. against Two Proteus mirabilis Isolated Clinical Strains in Different Anatomical Sites In Vitro: An Explorative Study to Improve the Therapeutic Approach"

_microorganisms, 2023, doi:10.3390/microorganisms11092201_

Round 1

Reviewer 1 Report

Comments and suggestions for authors

The manuscript describes a study based on an in vitro model that evaluates certain probiotic properties of Lactobacillus ssp isolated from buffalo milk, including L. casei, L rhamnosus, L. brevis, and two ATCC strains, one of Lactobacillus acidophilus and Lactiplantibacillus plantarum on support of PC3 prostatic epithelium cells and the T24 line of human bladder cells infected with P. mirabilis called bladder PMI and prostate epithelium PM2 that the authors previously described in other works.

It is necessary for the authors to develop a significant improvement in the manuscript to reinforce the focus of their work and highlight its relevance, because it is not entirely new. Also, it is important and essential that authors include a clearer description of various sections of their manuscript because they have some errors and inconsistencies throughout their manuscript that require attention.

Specific comments

Title: It is recommended that the authors analyze the title of their research because in the current presentation they do not reflect the research that the authors developed, possibly a more focused title attached to its main objective, which was the evaluation of probiotic strains of Lactobacillus ssp to Proteus mirabilis.

Abstract:

It is recommended that authors adhere to the guidelines of the journal for the preparation and improvement of their manuscript. In the summary, the authors present from line # 13 to line # 22 too much introduction making their research irrelevant to the readers and end with the objective of working in two lines (line # 23 and line # 24). This summary is not illustrative and should end with the main contribution of his study, I consider it should be structured.

Introduction:

The introduction should be rephrased in the final part of your research objective lines # 79 to line # 85 to highlight the intention of why this research is significant

Material and methods:

In the section on bacterial strains, PMI and PM2, line #92-10, it is important that the authors describe certain particular characteristics of these strains or cite some references, since they only describe the BLAST analysis cited as reference 30, a previous work by the authors.

In the biofilm formation essay section, on line # 112, of their manuscript, the authors include (O'Toole) and no reference number. I recommend authors homogenize their references if they are going to cite them by author or by consecutive number.

I also attach the reference, which does not appear in its corresponding section, and I recommend to the authors a careful review of their manuscript.

O'Toole G. A. (2011). Microtiter dish biofilm formation assay. Journal of visualized experiments : JoVE, (47), 2437. https://doi.org/10.3791/2437

It is important to describe all the original specifications of your equipment in a more detailed and clear way with the commercial reagent brand.

In the section on cell culture line #134- #136, why do the authors use this MOI of 100:1? or explain the support of this relationship possibly with a reference

In the analysis section of the expression of the flagellar gene of the P. mirabilis bacterium by real-time PCR, the authors not only determined this gene, but also the expression of cytokines that they describe in their table 1 line # 163 to # 164. I recommend modifying their subtitle to include these genes and maintain continuity in the reading of the manuscript.

In the internalization assay section, include a reference support for this assay and rephrase the efficiency calculation with a mathematical formula lines # 179-180

In the Elisa assay section line #183 to line # 185, clearly detail the specifications of these Kit extrapolation detection values with standard curve time of collection of the supernatants or include a reference.

Results

In the section on antibiofilm activity from Lactobacillus to P. mirabilis line # 197-198, rephrasing the term is apparently unclear. In this first result of their study, why did the authors not include a control in their graph of biofilms by Lactobacillus alone, since it is reported that some strains of Lactobacillus are biofilm-formers on abiotic surfaces? Although the authors comment that for this reason they count the CFU/ml of the Proteus mirabilis, in microcolonies this is not clear.

In your graph of figure 1b of count of microcolonies because in the count of CSF/ml, for the isolate of PM1 and PM2 lines # 226 to line # 237, if the interaction conditions were the same, lower values were observed for PM2 alone in comparison with PM1?

Conclusions

I recommend that the authors rephrase their conclusion because they do not reflect the response to their objective after this screening of probiotic strains, what advantages are there against infection by these isolates. The diversity of probiotic strain effects is well described. Line #597 to line #615.

I recommend that the authors should focus their conclusions on the main data of their study and also consider its limitations.

In the section of the contributions of the authors I suggest that the authors follow the guidelines of the journal supported by the instructions for authors. Line # 617 to #619.

Finally

In the references section

I recommend authors to adhere to the guidelines of the Journal, most of the cited references do not comply with the journal format, I suggest being careful in this section

Author Response

Dear Reviewer, thank you for the careful review of our manuscript and for having highlighted some critical issues to be eliminated in order to improve its quality. We hope our changes and answers are adequate to your requests.

Title: It is recommended that the authors analyze the title of their research because in the current presentation they do not reflect the research that the authors developed, possibly a more focused title attached to its main objective, which was the evaluation of probiotic strains of Lactobacillus ssp to Proteus mirabilis.

As requested, we added the term “Evaluation” to highlight the goal of our study

Abstract:

It is recommended that authors adhere to the guidelines of the journal for the preparation and improvement of their manuscript. In the summary, the authors present from line # 13 to line # 22 too much introduction making their research irrelevant to the readers and end with the objective of working in two lines (line # 23 and line # 24). This summary is not illustrative and should end with the main contribution of his study, I consider it should be structured.

 Done

Introduction:

The introduction should be rephrased in the final part of your research objective lines # 79 to line # 85 to highlight the intention of why this research is significant

 Done.

Material and methods:

 In the section on bacterial strains, PMI and PM2, line #92-10, it is important that the authors describe certain particular characteristics of these strains or cite some references, since they only describe the BLAST analysis cited as reference 30, a previous work by the authors.

We have added to the materials and methods section some lines in which we briefly list, with the appropriate references, the other analyzes that have been carried out to characterize the two strains

In the biofilm formation essay section, on line # 112, of their manuscript, the authors include (O'Toole) and no reference number. I recommend authors homogenize their references if they are going to cite them by author or by consecutive number.

 Done

It is important to describe all the original specifications of your equipment in a more detailed and clear way with the commercial reagent brand.

 Done

 In the section on cell culture line #134- #136, why do the authors use this MOI of 100:1? or explain the support of this relationship possibly with a reference.

 The 100:1 ratio is widely used in microbiology; however, we have added a series of references as required.

 In the analysis section of the expression of the flagellar gene of the P. mirabilis bacterium by real-time PCR, the authors not only determined this gene, but also the expression of cytokines that they describe in their table 1 line # 163 to # 164. I recommend modifying their subtitle to include these genes and maintain continuity in the reading of the manuscript.

By our mistake, table 1 with the primer sequence list was placed after the Real-Time PCR paragraph; it went after the flhDC gene analysis paragraph, as that was where it was first mentioned. We have therefore relocated the table; however, we cannot make a single paragraph on flhDC and cytokines as they are two different experiments.

In the internalization assay section, include a reference support for this assay and rephrase the efficiency calculation with a mathematical formula line # 179-180.

 Done

In the Elisa assay section line #183 to line # 185, clearly detail the specifications of these Kit extrapolation detection values with standard curve time of collection of the supernatants or include a reference.

 More details on the procedure have been added to the paragraph. The results have been normalized based on the standard curves of each kit according to the manufacturer’s instructions.

 Results:

 In the section on antibiofilm activity from Lactobacillus tP. mirabilis line # 197-198, rephrasing the term is apparently unclear. In this first result of their study, why did the authors not include a control in their graph of biofilms by Lactobacillus alone, since it is reported that some strains of Lactobacillus are biofilm-formers on abiotic surfaces? Although the authors comment that for this reason they count the CFU/ml of the Proteus mirabilis, in microcolonies this is not clear.

 The lactobacilli alone as controls were not used because, having already analyzed two pathogenic strains and 6 strains of lactobacilli, inserting the controls too would have been too confusing both to analyze (at that point we would have had to subtract the values obtained from each single lactobacillus from the values obtained from each single pathogen-probiotic interaction) and above all to be represented on a graph without confusing the reader. For this reason, we thought it more appropriate and clearer to count the CFUs/ml of PM1 and PM2 only, using a selective medium on which lactobacilli could never grow, to have the effective data of their activity. However, the sentence has been rewritten to make it clearer.

In your graph of figure 1b of count of microcolonies because in the count of CSF/ml, for the isolate of PM1 and PM2 lines # 226 to line # 237, if the interaction conditions were the same, lower values were observed for PM2 alone in comparison with PM1?

 The interactions are the same, but the behavior of the PM1 and PM2 strains in the biofilm formation assay changes; in fact, as demonstrated in our previous work in which we analyzed all the differences between the two clinical isolates (Ref. 30), PM2 forms less biofilm than PM1, causing an acute infection with a strong inflammatory state, while PM1 forms a more abundant biofilm and causes chronic infection with weaker and slower proinflammatory cytokine release.

Conclusions:

 I recommend that the authors rephrase their conclusion because they do not reflect the response to their objective after this screening of probiotic strains, what advantages are there against infection by these isolates. The diversity of probiotic strain effects is well described. Line #597 to line #615.

I recommend that the authors should focus their conclusions on the main data of their study and also consider its limitations.

 Done

 In the section of the contributions of the authors I suggest that the authors follow the guidelines of the journal supported by the instructions for authors. Line # 617 to #619.

 Done

 Finally

In the references section

I recommend authors to adhere to the guidelines of the Journal, most of the cited references do not comply with the journal format, I suggest being careful in this section

 Done

The recommended corrections were all performed and highlited in the text.

The language editing was performed by papertrue.com.

            We hope that the revised manuscript is now suitable for publication.

Best regards,

Giovanna Donnarumma

Alessandra Fusco

Reviewer 2 Report

The manuscript "Different activity of Lactobacillus spp. against two Proteus mirabilis isolated clinical strains in different anatomical sites in vitro: an explorative study to improve the therapeutic approach" by Fusco et al. describes the use of Lactobacillus strains as an alternative to antibiotics to prevent the growth of uropathogenic Proteus mirabilis.  This is a topical manuscript as several recent studies have highlighted the fact that Lactobacillus are negatively associated with many urologic diseases and that bacteria from this genus can help to prevent UTIs.  The manuscript can be improved through the following changes:

"[Lactobacillus] are also frequently endowed with antibiotic resistance, which is useful in the case of combination therapies with antibiotics [22, 23]."

Several recent publications on biofilms attached to extracted urologic implants contradict this statement.  In those studies, Lactobacillus levels were lower and uropathogen levels higher in patients that received antibiotics in the preceding 30 days.  See pmid's (37460611, 36724057, 36672723).

Authors suggest that Lactobacillus strains would be clinically active (prevent P. mirabilis infections) if taken as an oral probiotic.  However, all assays were conducted with direct interactions with Lactobacillus.  Thus, in order for the probiotic idea to work, Lactobacillus would have to make it into the gut, pass through the epithelial barrier into circulation, get filtered through the kidneys and into the proximate location where Proteus causes infection.  This is neither probable nor desirable. Previous clinical trials that examined the efficacy of L. crispatus used vaginal inoculations of the strain to prevent UTIs - putting the probiotic next to the infectious bacteria.  See pmids 36949381, 21498386, 34258813). 

"In recent years, we have characterized two clinically isolated strains of P. mirabilis, which we named PM2 and PM2" - PM2 is listed twice

Although not critical for this study - Have there been no efforts to acquire additional P. mirabilis strains since 2009?  The study would be much more robust if additional isolates were available and I think it would be easy enough to obtain them.

Similarly, why choose Lactobacillus strains from buffalo milk?  Lactobacillus is a very common genus in the urinary tract of men and women and can be easily isolated from the urine.  Such clinically relevant isolates, I think, would be more meaningful in this setting and more applicable to a patient population.

Author Response

Dear Reviewer, thank you for the suggestions you have provided. Here are our answers. The suggested changes are highlighted in the manuscript, we hope we have satisfied your requests and doubts.

"[Lactobacillus] are also frequently endowed with antibiotic resistance, which is useful in the case of combination therapies with antibiotics [22, 23]."

 Several recent publications on biofilms attached to extracted urologic implants contradict this statement.  In those studies, Lactobacillus levels were lower and uropathogen levels higher in patients that received antibiotics in the preceding 30 days.  See pmid's (37460611, 36724057, 36672723).

 According with your suggestions, we removed the sentence.

Authors suggest that Lactobacillus strains would be clinically active (prevent P. mirabilis infections) if taken as an oral probiotic.  However, all assays were conducted with direct interactions with Lactobacillus.  Thus, in order for the probiotic idea to work, Lactobacillus would have to make it into the gut, pass through the epithelial barrier into circulation, get filtered through the kidneys and into the proximate location where Proteus causes infection.  This is neither probable nor desirable. Previous clinical trials that examined the efficacy of L. crispatus used vaginal inoculations of the strain to prevent UTIs - putting the probiotic next to the infectious bacteria.  See pmids 36949381, 21498386, 34258813). 

 In truth, the manuscript does not really speak of oral intake but of manipulation of the intestinal microbiota; oral administration is the most convenient for achieving this goal, but we do not even question its limitations as regards reaching the destination sites (urinary tract). On the other hand, it is true that the benefits that the intake of probiotics can bring to the intestinal microbiota also affect the translocation of uropathogens in the urinary tract, and that direct interaction between the probiotic and pathogen can already take place in the gut, being P. mirabilis an enterobacterium normally found in the intestinal tract. Topical administration, as in the case of vaginal ovules, is certainly very convenient for local applications, as it can ensure rapid, efficient colonization and a high local bacterial load; on the other hand, we have cited a recent work (Ref. 51) in which clinical studies on the daily intake of L. casei capsules are giving promising results in the treatment of bacterial prostatitis, in combination with antibiotic therapy. However, we have modified the manuscript by citing the suggested references (lines 610-611) and added this criticism in the conclusions section (lines 625-627) by counting it in the limitations of our study and in the future prospects to be investigated.

 "In recent years, we have characterized two clinically isolated strains of P. mirabilis, which we named PM2 and PM2" - PM2 is listed twice

The typo has been corrected.

 Although not critical for this study - Have there been no efforts to acquire additional P. mirabilis strains since 2009?  The study would be much more robust if additional isolates were available, and I think it would be easy enough to obtain them.

You are right, but we did not have other clinical isolates characterized in detail such as PM1 and PM2 (Ref. 5 and 30), furthermore these two strains continue to be very interesting as they have completely different, sometimes opposite, characteristics and behavior, so they seemed to us a good reference to analyze the behavior of lactobacilli under different infection conditions.

Similarly, why choose Lactobacillus strains from buffalo milk?  Lactobacillus is a very common genus in the urinary tract of men and women and can be easily isolated from the urine.  Such clinically relevant isolates, I think, would be more meaningful in this setting and more applicable to a patient population.

 The lactobacilli used were chosen according to the objectives of the project Incube-MISE Sportello Prog. N. F/200035/01/X45 (indicated at the end of the manuscript) which financed our work, and in which R&D—IBSA Farmaceutici Italia, the manufacturer of the probiotic strains, is the project leader; in fact, other papers have already been published on the use of these strains against other bacteria and on other epithelia (Ref. 29, 34 and 49). Moreover, since these are strains commonly used in the formulation of pharmacological preparations, we did not consider that the source of isolation could represent a significant difference.

The recommended corrections were all performed and highlited in the text.

The language editing was performed by papertrue.com.

            We hope that the revised manuscript is now suitable for publication.

Best regards,

Giovanna Donnarumma

Alessandra Fusco

Round 2

Reviewer 1 Report

The authors satisfactorily responded to alll the observations suggested by the reviewer for the improvement of their manuscrpt in this new version